# Effects of Ultrasound-Assisted Immersion Freezing on the Protein Structure, Physicochemical Properties and Muscle Quality of the Bay Scallop (*Argopecten irradians*) during Frozen Storage

**DOI:** 10.3390/foods11203247

**Published:** 2022-10-18

**Authors:** Bing Liu, You-Lin Liao, Liang-Liang Jiang, Miao-Miao Chen, Shan-Bin Yang

**Affiliations:** 1Engineering Research Center of Active Substance and Biotechnology, Ministry of Education, College of Chemistry, Chongqing Normal University, Chongqing 401331, China; 2School of Geography and Tourism, Chongqing Normal University, Chongqing 401331, China

**Keywords:** scallop, ultrasound-assisted immersion freezing, myofibrillar proteins, physicochemical properties, frozen storage

## Abstract

In this study, the comparison effects of ultrasound-assisted immersion freezing (UIF) at different ultrasonic power, immersion freezing (IF), and air freezing (AF) on the protein thermal stability, protein structure, and physicochemical properties of adductor muscle of scallop (*Argopecten irradians*) (AMS) during frozen storage were investigated. Principal component analysis and the Taylor diagram were used to comprehensively analyze all the indicators tested. The results showed that the UIF at 150 W (UIF-150) treatment was the most effective way to delay the quality deterioration of AMS during 90-day frozen storage. This was mainly because, compared to AF and IF treatments, UIF-150 treatment more effectively minimized the changes in the primary, secondary and tertiary structures of myofibrillar proteins, and it preserved the protein thermal stability of AMS by producing small and regular ice crystals in the AMS tissue during the freezing process. Moreover, the results of physicochemical properties indicated that UIF-150 treatment significantly inhibited the fat oxidation and microbiological activities of frozen AMS, and it finally maintained the microstructure and texture of AMS during frozen storage. Overall, UIF-150 has potential industrial application prospects in the rapid freezing and quality preservation of scallops.

## 1. Introduction

Scallops are widely accepted for their delicious taste and desirable nutrition, such as vitamins, n-3 polyunsaturated fatty acids, bioactive peptides and other bioactive compounds [1,2]. However, they are highly perishable in their fresh state, giving them a limited shelf life. Therefore, some preservation techniques and methods have been applied to guarantee the safety and quality of scallop products, including drying [1,3], cooling [4,5], freezing [6,7], and so on. Among these, freezing and frozen storage, a commonly used preservation technique, is widely used to extend the shelf life of scallops and scallop products [7]. However, initial chemical reactions, ice crystals, endogenous proteolysis as well as microbial growth can lead to the adductor muscle of the scallop (AMS, the main edible part) spoilage and quality deterioration during the freezing process and long-term frozen storage. AMS finally suffers from poor taste and texture, which seriously affects the commercial value of scallop products [6,7,8]. Therefore, exploring suitable freezing methods to maintain the quality of frozen scallop products is crucial.

At present, traditional freezing techniques, such as air blowing, immersion freezing (IF), and plate contact, are widely used in food factories [9,10,11]. However, previous studies indicated that the uneven and large ice crystals produced using traditional air freezing (AF) methods adversely affect product quality, especially protein-rich foods [9,10,12,13]. Specifically, large ice crystals are formed and aligned both inside and outside the cell during the freezing and frozen storage period, disrupting the myofibrils and connective tissue of muscle tissue. This leads to aggregation, cross-linking, and conformational changes in muscle proteins, or possibly destructing their secondary and tertiary structures, finally resulting in the irreversible denaturation in structural proteins and subsequent quality deterioration of muscle tissue, thereby affecting product quality [10,12,14,15].

Ultrasonic-assisted immersion freezing (UIF), as a novel green freezing technology, has no legislative difficulties and no chemical pollution and has been widely used in meat factories [10,12,14,16]. Ultrasonic waves can exert mechanical cavitation and thermal effects. Under these effects, the ice crystals formed during the freezing of meat products are small and regular, thereby reducing the quality loss caused by ice crystals to the frozen meat products [10,17]. In addition, during frozen storage, protein-rich muscle products are prone to protein denaturation, fat oxidation and spoilage caused by endogenous proteases and microorganisms, which ultimately lead to poor product quality [16,18]. Numerous reports indicated that UIF could effectively inhibit the quality deterioration of meat products during frozen storage by inhibiting the aforementioned adverse reactions [10,14,16,19]. For example, Zhang et al. [16] found that the ultrasound-assisted immersion freezing could effectively inhibit the quality deterioration of chicken breast during frozen storage by retarding the protein thermal stability loss, protein structure degradation, and protein oxidation of muscle tissue. Ma et al. [19] reported that the multi-frequency ultrasound-assisted freezing at 175 W could effectively improve the physicochemical quality of Cultured Large Yellow Croaker (*Larimichthys crocea*) by enhancing the freezing rate and inhibiting the protein degradation caused by ice crystals. However, few reports exist on the application of UIF technology to preserve the quality of frozen scallop products.

Myofibrillar proteins (MPs) are the major proteins of AMS (60–70% of the total protein content) [20]. Any physicochemical and functional changes in MPs during storage are closely linked to the quality of AMS [4,20]. However, to the best of our knowledge, the mechanism of quality deterioration of AMS that relates to structural proteins degradation during frozen storage is still in need of clarification. Thus, the goal of present work was to investigate the comparison effects of UIF at different ultrasonic power, IF, and AF on the quality of adductor muscle of scallop (*Argopecten irradians*) (AMS) during frozen storage and to unravel the action mechanism involved. First, the primary, secondary, and tertiary structural changes in MPs from frozen stored AMS were analyzed by determining the sodium dodecyl sulfate–polyacrylamide gel electrophoresis (SDS-PAGE), fourier transform infrared spectroscopy (FTIR), and fluorescence intensity of MP samples. Furthermore, the protein thermal stability, microstructure, fat oxidation, spoilage, thawing loss, and texture properties of the muscle tissue were analyzed by testing the differential scanning calorimetry (DSC), scanning electron microscopy (SEM), thiobarbituric acid reactive substance (TBARS), and total volatile base nitrogen (TVB-N) of AMS samples. Finally, the multivariate data analysis, including principal component analysis (PCA) and Taylor diagram, was performed to quickly select the best freezing conditions and provide a relationship between protein thermal stability, protein structure, and all physicochemical properties. 

## 2. Materials and Methods 

### 2.1. Materials and Chemicals

All live scallops were obtained from a local market in Chongqing, China, stored in an ice-filled incubator, and shipped to the laboratory immediately after purchase. The AMSs were manually stripped from the shell using a spatula at 4.0 ± 1.0 °C in an ice water bath (water, ice and NaCl (1:5:1, *w*/*w*/*w*)). The obtained AMSs were rinsed with water and stored at 4 °C refrigerator until treatment. All chemicals were of analytical grade and purchased from Chongqing Chuandong Reagent Co., Ltd. (Chongqing, China).

### 2.2. Preparation of Muscle Samples 

The process design of all experiments in this study is shown in Figure 1. The study included eight treatments; approximately 0.5 kg of AMS was used for the analysis in each treatment. Three different freezing processes (AF, IF, and UIF) were used to freeze the AMS samples. AF group: a normal freezer at −18.0 ± 0.5 °C was used to freeze samples. IF and UIF groups: as shown in Figure 2, a new ultrasonic-assisted immersion freezer (Chongqing Jianke Co., Ltd., Chongqing, China) using 95% ethanol and 5% fluoride as coolants was applied to freeze samples. As shown in Figure 2B, the advantage of this device was that it has a lever (15) that can be turned. In brief, motor (11) provides power to the connecting rod (12); then, the connecting rod rotates by driving the lever (15) through the rotating plate (16). Thus, the AMSs inside could kept rolling, effectively solving the problems of incomplete freezing and the long freezing time of raw materials during the freezing process. The temperature of coolants was always maintained at −18 ± 0.5 °C. The ultrasonic parameters in this study were set according to previous studies [14,16]. An ultrasonic frequency of 30 kHz and a cycle of 30 s on/30 s off ultrasonic intermittent mode were set with a Φ20 mm ultrasonic probe. The ultrasound power was set at 0, 100, 125, 150, 175, and 200 W, named IF, UIF-100, UIF-125, UIF-150, UIF-175, and UIF-200, respectively. The whole freezing process was completed when the center temperature of the sample reached about −18 °C, which takes about 10 min. Then, the samples from all groups were continuously frozen in a common refrigerator for 90 days at a temperature of −18 ± 1 °C.

### 2.3. Extraction of MPs

MPs were obtained according to previous study [20]. Briefly, minced AMS was homogenized with four times the volume (*v*/*w*) of phosphate buffer (0.1 M NaCl, 1 mM EDTA and 2 mM MgCl_2_, pH = 7.0). Then, the mixture was sieved through a sieve and centrifuged at 5000× *g* for 10 min at 4 °C. The sediment was collected. This process was repeated two times. Lastly, the sediments were collected and freeze-dried. The freeze-dried MP samples were stored in a −30 °C freezer for analysis as soon as possible.

### 2.4. SDS-PAGE 

Sodium dodecyl sulfate–polyacrylamide gel electrophoresis (SDS-PAGE) of MP samples was performed as described in our previous study [20]. The electrophoresis loading volume was 10 μL, the concentration of stacking gel was 5%, and the concentration of the resolving gel was 8%. After staining with Coomassie brilliant blue R-250, the gel pieces were destained overnight. Image J 1.37v software was used to analyze the images with SDS-PAGE.

### 2.5. FTIR 

Fourier transform infrared spectroscopy (FTIR) of the MP samples was performed by a previously described method [21] using a Nicolet iS5e FTIR spectrometer (Thermo Fisher Scientific, Waltham, MA, USA). In brief, the spectra were collected from 32 cumulative scans in the 4000 to 400 wavenumber range with a resolution of 4 cm^−1^. The PeakFit software (version 4.04, SPSS Inc., IL, USA) was used to perform the fitting on the amide I band of MP (1600–1700 cm^−1^). 

### 2.6. Fluorescence Intensity

The fluorescence intensity of MP samples was assayed according to a previously described method [22] using a Perkin-Elmer LS fluorescence spectrophotometer (Perkin-Elmer Corp., Waltham, MA, USA). In short, the MP solution was diluted to a concentration of 0.4 mg/mL with 15 mM phosphate-buffered saline buffer (pH = 6.25) containing 0.6 M NaCl. The excitation wavelength and scan wavelength were set to 283 nm and 300–400 nm, respectively.

### 2.7. DSC

The differential scanning calorimetry (DSC) of AMS samples was performed as described in a previous study [4] using a Perkin-Elmer 8000 DSC (Perkin-Elmer instruments, Waltham, MA, USA). About 200 mg of AMS tissue was weighed, and an equal amount of distilled water was used as the blank control. The heating rate was 1 °C/min, and the heating range was 5–80 °C.

### 2.8. Scanning Electron Microscopy Observation

The scanning electron microscopy (SEM) of AMS samples was performed using a scanning electron micrograph (JSM-7800F, Japan Electron Optics Laboratory, Tokyo, Japan). Briefly, the AMS tissues were dehydrated with 50, 60, 70, 80, 90, and 100% ethanol. Then, the freeze-dried samples were imaged and operated at 3–10 kV. 

### 2.9. Determination of TBARS and TVB-N

The TBARS of the AMS samples was determined as described in previous study [4]. In short, 0.5 g of AMS powder was mixed with 2 mL of distilled water and 2 mL of trichloroacetic acid (10%, *w*/*v*) solution. After vortexing for 2 min, the mixture was centrifuged at 8000× *g* for 5 min. Thereafter, 1 mL of the supernatant was blended with 1 mL of thiobarbituric acid (0.01 M) and reacted in boiling water for 25 min. Then, the absorbance of the mixture was determined at 532 nm. The results of TBARS were presented as mg malondialdehyde (MDA)/g dry adductor muscle.

The total volatile base nitrogen (TVB-N) of AMS samples was determined as described in our previous study [4]. The method of Conway microdiffusion was used. The TVB content expressed in mg N/100 g AMS.

### 2.10. Determination of Thawing Loss and Texture

The thawing loss of the AMS samples was measured according to Sun et al. [12]. In brief, frozen AMS was weighed immediately before (*W*_0_) and after (*W*_1_) thawing. The thawing loss was calculated using the following formula: $$Thawing\;loss\left(\%\right)=\frac{W_{0}-W_{1}}{W_{0}}$$

The texture profile analysis (TPA) of AMS samples was performed as previously described [20] using a JS-4Pro+ texture analyzer (Chuangxing Electronic Equipment Manufacturing Co., Ltd., Tianjin, China). In brief, each AMS was trimmed into a cylindrical shape. The test conditions were set to 60% of a compression level and 1.0 mm/min of constant probe speed using a P/100 probe. Ten replicates were determined for each group.

### 2.11. Statistical Analysis

Most data in the study were analyzed using an SPSS environment. The effective ultrasound treatment was selected using the Taylor diagram, which was performed in the R package (“openair”). The statistical differences between samples were determined using Duncan’s multiple range and analysis of variance tests. The significance was judged at the 95% level (*p* < 0.05). All experiments were repeated three times.

## 3. Results and Discussion

### 3.1. Changes in the Primary Structure of MPs

The SDS-PAGE of MPs was investigated from control AMS and 90-day frozen stored AMS with different treatments, as shown in Figure 3A. Actin (43–48 kDa) and myosin heavy chain (MHC, 224–228 kDa) bands were quantified in Appendix A. The intensity of MHC bands in 90-day frozen stored samples was lower than that in the control sample, indicating that the proteolysis of MPs occurred during frozen storage [13,16]. Specifically, after 90 days of frozen storage, the intensity of MHC bands in the UIF 100–200, IF, and AF samples declined by 27.9%, 5.2%, 5.8%, 9.7%, 22.7%, 5.8%, and 7.8% (*p* < 0.05). In contrast, the treatment with UIF-125, UIF-150 and IF could effectively inhibit the degradation of MHC caused by the 90-day frozen storage. Meanwhile, compared with the frozen stored samples of the other groups, no obvious loss of actin bands was observed in the samples of UIF-125 and UIF-150, indicating that the treatments of UIF-125 and UIF-150 could effectively inhibit the degradation of actin caused by the 90-day frozen storage. The above results showed that the UIF-125 and UIF-150 treatment significantly retarded the primary structural degradation of MPs. This may be attributed to the collapse of cavitation bubbles produced by UIF-125 and UIF-150, which broke large ice crystals into small fragments and reduced the mechanical damage to MPs during frozen storage [16]. Similarly, earlier studies also reported that suitable ultrasonic treatment could effectively inhibit the degradation of fish MPs during frozen storage [10,12,23].

### 3.2. Changes in the Secondary and Tertiary Structure of MPs

The changes in the secondary structure of MPs from control AMS and 90-day frozen stored AMS with different treatments were detected using FTIR. The 90-day frozen storage resulted in a decreased amide I peak ratio, as shown in Figure 3B. The amide I band (1600–1700 cm^−1^) was studied as a deconvoluted spectrum to further visually understand the changes in MP secondary structure, including random coil, α-helix, β-sheet, and β-turn, and the corresponding data are shown in Figure 3C. The 90-day frozen stored samples showed an increase in random coil and β-turn values and a decrease in α-helix values compared with the control samples, which was consistent with previous findings [12,24,25,26]. This might be because the formation of ice crystals and proteolysis during frozen storage resulted in the destruction of the secondary structure of MP [24,27]. For example, after 90-day frozen storage, the proportion of α-helix in AF, IF, and UIF 100–200 samples decreased by 73.3%, 59.2%, 62.9%, 47.4%, 41.5%, 62.6%, and 68.8% (*p* < 0.05). Particularly, the UIF-150 treatment significantly retarded the degradation of the secondary structure of MPs.

The changes in the tertiary structure of MPs in samples under different treatments were analyzed with fluorescence spectroscopy by measuring their inherent fluorescence emission [28]. Figure 4A shows that 90-day frozen storage resulted in a significant reduction in the fluorescence intensity and a bathochromic shift of *λ*_max_ from 332.2 nm (control) to 333.8 nm (AF). These results indicated that the partially buried tryptophan residues (the sample chromophore) in MPs were exposed to a polar environment after 90-day frozen storage [14,28]. In contrast, the *λ*_max_ of the UIF-150 sample was lower than that of the other samples (*p* < 0.05), which was possibly because the UIF treatment at 150 W accelerated the freezing process and inhibited the unfolding of protein structures. Subsequently, the damage to the tertiary structure of the MP was reduced. Zhang et al. [16] also reported that the λ_max_ redshift and the decrease in fluorescence intensity in the MP samples indicated that the integrity of the tertiary structure of MPs was disrupted by the growth of ice crystals. UIF treatment can effectively retard these structural changes during long-term frozen storage, thereby inhibiting the quality deterioration of frozen chicken breasts. 

The results of SDS-PAGE, FTIR and fluorescence spectroscopy suggested that the UIF-150 treatment retarded the structural degradation of MPs caused by the freezing process and long periods of frozen storage. In summary, these results were attributed to two reasons: (1) the new instrument shortens the freezing process, (2) the cavitation bubbles generated by the UIF-150 treatment were speculated to form regular-shaped and small-sized ice crystals in AMS, thereby reducing the destruction to the MP structure by the ice crystals during long-term frozen storage. Qiu et al. [10] also found that UIF treatment could increase the heat-transfer coefficient and significantly improve the freezing rate of samples. The more rapid the freezing rate, the smaller and more uniform the formation of ice crystals and the lesser the mechanical damage to MPs during frozen storage. Earlier studies similarly reported that the structural degradation of MPs from fish and pork caused by ice crystals was also inhibited by the UIF treatment [9,14]. 

### 3.3. Changes in the Thermal Denaturation of AMS

DSC is a convenient and commonly used technique for determining the temperature of protein denaturation [29,30]. Figure 4B and Appendix A show that all samples in the control group and the frozen groups with different treatments showed two characteristic peaks at 45.81–48.19 °C (*T*_max1_) and 67.10–72.26 °C (*T*_max2_), representing the denaturation temperatures of myosin and actin, respectively [31]. The *T*_max_ value of the frozen stored AMS decreased to lower temperatures after 90 days of frozen storage compared with control AMS. These results indicated that long periods of frozen storage reduced the thermal stability of structural proteins in AMS possibly because of the denaturation of actin and myosin caused by proteolysis and ice crystals [14]. Specifically, the *T*_max1_ of AMS in UIF 100–200, IF, and AF samples decreased by 2.1%, 4.1%, 3.9%, 4.9%, 4.1%, 4.2%, and 4.9% after 90 days of frozen storage (*p* < 0.05). The *T*_max2_ of AMS in UIF 100–200, IF, and AF samples decreased by 7.1%, 2.9%, 1.7%, 2.8%, 6.9%, 4.4%, and 5.6% after 90-day frozen storage (*p* < 0.05). Obviously, UIF-100 and UIF-150 treatments significantly inhibited *T*_max1_, and *T*_max2_ decreased (*p* < 0.05), respectively. Similarly, the denaturation enthalpies of actin and myosin, Δ*H*_1_ and Δ*H*_2_, in UIF 100–200, IF and AF groups also reduced significantly after 90-day frozen storage (*p* < 0.05). Earlier studies revealed that the protein denaturation triggered by the frozen storage process resulted in increased protein aggregation and the weakening of hydrogen bonds, resulting in the loss of Δ*H* [9]. In contrast, the Δ*H*_1_ and Δ*H*_2_ values of the UIF-150 samples (0.63 and 0.39 J/g, respectively) were higher than those of the other groups. This indicated that the degree of protein denaturation in UIF-150-treated samples was significantly lower than that in other groups. The DSC results further confirmed the analysis of SDS-PAGE, FTIR and fluorescence spectroscopy, indicating that the UIF-150 treatment could reduce the structural protein degradation of AMS.

### 3.4. Changes in the Microstructure of AMSs 

The SEM results of control AMS and AMS in different treatment groups after 90 days of frozen storage are displayed in Figure 5. The 5000× microscopic image of the control sample showed that the myofibrils were tightly packed without interfibrillar spaces between them. A high-power microscopic image (20,000×) showed that the honeycomb structure of connective tissues was tethered around the myofibrils [2,4]. The AMS samples after 90-day frozen storage showed significant changes in their microstructure. For example, in the AF sample, the connective tissue almost disappeared, the honeycomb-like network structure was no longer visible, a large number of aggregates appeared at the same time, and large gaps were also created between the myofibrils. This might be due to the effects of ice crystals and proteolysis, resulting in the degradation of MPs and connective tissue proteins [2,11]. However, the microstructure of the AMS sample under the appropriate ultrasonic freezing conditions had different morphologies. In contrast, it was seen that in the UIF-150 samples, a small part of the network structure of connective tissue could still be observed, and no obvious gaps were found between the myofibrils. These results indicated that in the UIF-150 samples, the structural protein degradation was reduced and the microstructure of AMS was maintained. 

The results of SEM confirmed that the protein structure damage and protein denaturation degree of the UIF-150 treatment samples were lower than those in other treatment groups, which were consistent with the above results of SDS-PAGE, FTIR, fluorescence spectroscopy and DSC.

### 3.5. Changes in the TBARS and TVB-N of AMS

TBARS is an important indicator of fat peroxidation [32]. The higher the TBARS value of the sample, the higher the degree of fat oxidation. In this study, compared with the control sample, the TBARS values (Table 1) in 90-day frozen samples increased significantly, indicating that fat oxidation occurred in AMS tissues after 90-day frozen storage. Specifically, compared with the control sample, the AMS from the AF, IF, and UIF 100–200 treatment groups showed a 2.21-, 0.84-, 1.29-, 0.78-, 0.49-, 0.86-, and 1.11-fold increase in the TBARS content. Obviously, the UIF-150 treatment could effectively delay the fat oxidation caused by frozen storage. This might be because the UIF-150 treatments reduced the activity of lipase, which caused fat oxidation. Our previous study found that the ultrasound treatment significantly inhibited the lipase activity of AMS (*p* < 0.05), and the possible mechanism was that the chemical and mechanical impacts were generated by ultrasonic cavitation, which could inactivate the enzymes [4]. 

The changes in the quality of aquatic products during storage were investigated by TVB-N analysis, which was widely used in previous studies [12,19,26,33,34]. The TVB-N content of AMS after 90-day frozen storage increased significantly (Table 1). Specifically, the TVB-N values of AMS in UIF 100–200, IF, and AF groups showed a 2.75-, 2.20-, 1.80-, 2.53-, 2.90-, 2.51-, and 3.06-fold increase compared with the control sample. The UIF-150 treatment suppressed the increased TVB-N value of AMS, which might be attributed to the fact that UIF at 150 W could effectively inactivate microorganisms on the surface of AMS, thereby delaying the frozen shelf life of scallops. Similarly, Sun et al. [12] reported that the UIF treatment inhibited the microbiological activities of frozen fish during 180 days of storage.

The aforementioned results indicated that the UIF-150 treatment significantly inhibited fat oxidation and spoilage in AMS during frozen storage.

### 3.6. Changes in Thawing Loss and Texture of AMS

Thawing loss and texture are critical indicators of the frozen scallop quality [18,34]. As shown in Table 1, 90-day frozen storage resulted in a significant increase in the thawing loss of AMS, which was possibly because the growth of ice crystals over frozen storage disrupted the tightly packed muscle tissue by squeezing adjacent muscle fibers. Compared with the control sample, the thawing losses of frozen stored AMS in UIF 100–200, IF, and AF groups increased by 2.91-, 2.71, 2.07-, 2.94-, 2.84-, 2.79-, and 3.04-fold (*p* < 0.05). Notably, the UIF-150 treatment significantly retarded the decrease in thawing loss in AMS after 90-day frozen storage. This was attributed to the cavitation bubbles generated by the UIF-150 treatment, which were speculated to form regular-shaped and small-sized ice crystals in AMS during the UIF process, thereby reducing the destruction to the MP structure and AMS microstructure by the ice crystals during long-term frozen storage. Similarly, Sun et al. [12] found that the decrease in the thawing loss of frozen fish was inhibited significantly by the treatment of UIF.

As shown in Table 1, all of the tested textural parameters of AMS significantly decreased after 90 days of frozen storage. Previous studies reported that the texture of aquatic muscle foods became softer after long-term frozen storage, which might be attributed to the proteolytic and microstructural damage that occurred during frozen storage [12,35,36]. Specifically, the samples in the AF, IF, and UIF 100–200 groups showed a 28.6%, 17.7%, 26.3%, 17.9%, 11.8%, 21.4%, and 24.7% decrease in hardness, 21.2%, 15.2%, 24.2%, 18.2%, 10.6%, 19.7%, and 21.2% decrease in springiness, and 36.5%, 34.4%, 25.9%, 19.6%, 13.3%, 28.0%, and 25.6% decrease in chewiness compared with the control sample. Obviously, the UIF-150 treatments retarded the texture deterioration of AMS caused by frozen storage. Based on the findings of the previous sections, this was because the use of a new instrument and the cavitation generated by UIF-150 treatments accelerated the freezing speed of AMS, promoted the generation of uniform and small ice crystals, and inhibited the protein denaturation, fat oxidation and spoilage of AMS during frozen storage. Then, the decreased thawing loss of AMS could maintain the microstructure and texture of scallops. Similarly, earlier studies have reported that suitable UIF treatment could effectively retard the quality deterioration of frozen muscle products [12,14,37].

### 3.7. Selection of an Effective Ultrasound Treatment

A Taylor diagram could quickly calculate the degree of difference between groups by comparing the centered RMS difference and relationship [4]. In this study, all the detected indicators, including SDS-PAGE, FTIR, fluorescence intensity, DSC, TBARS, TVB-N, thawing loss and textural properties, were analyzed by using the Taylor diagram to quickly and accurately determine the most effective ultrasound treatment conditions. The control group in this study was used as a reference, and the differences between the other groups and the control group were analyzed by comparing their correlation and centered RMS difference using the Taylor diagram. 

As shown in Figure 6A, the correlation coefficient between the frozen groups and the control group was represented by the azimuthal positions, while the central RMS difference was represented by the distance between them. The results showed that the correlation coefficient between the frozen groups and the control group from high to low was as follows: UIF-150 > UIF-125 > UIF-175 > IF > UIF-200 > UIF-100 ≥ AF, and the central RMS difference between the frozen groups and the control group from low to high was as follows: UIF-150 < UIF-125 < IF < UIF-175 < UIF-200 < UIF-100 < AF. Clearly, the UIF-150 group had the highest correlation coefficient and the smallest central RMS difference value with the control group, indicating that the UIF-150 samples have the best quality. In summary, according to the Taylor diagram results, the degree of quality deterioration in each group of samples after 90-day frozen storage from low to high was as follows: UIF-150 < UIF-125 < UIF-175 < IF < UIF-200 < UIF-100 < AF. Overall, UIF-150 treatment could effectively postpone the quality deterioration of AMS caused by long-term frozen storage, but the UIF treatment with too low, and too high ultrasonic waves could not achieve the desired effect.

### 3.8. Relationship between All Physicochemical Properties

The aforementioned results showed that freezing conditions played a critical role in the physical properties of AMS. PCA was applied to obtain a linear combination of the primary, secondary, and tertiary structures of MPs, protein thermal stability, microstructure, fat oxidation, texture properties, and quality between the control sample and 90-day frozen stored samples under different treatments.

The PCA bi-plot diagram in Figure 6B displays both observations and variables in a new space. Here, 88.34% of the total variation was explained by the first two principal components, which accounted for 59.52% and 28.82%, respectively. As shown in Appendix A, Δ*H*_1_, α-helix, *T*_max1_, thawing loss, TVB-N, random coil, hardness, chewiness, Δ*H*_2_, TBARS, and springiness were identified in the main variables for PC1. Among them, thawing loss, TVB-N, random coil, and TBARS negatively correlated with other parameters. The variables for PC2 were MHC, actin, *T*_max2_, β-turn and β-sheet. Among them, *T*_max2_ and β-sheet negatively correlated with all three other parameters. A close correlation existed between the freezing conditions, protein structure, protein thermal stability, textural properties, fat oxidation, and quality in AMS. In addition, thawing loss, TBARS, TVB-N, hardness, chewiness, and springiness were quite closely linked to the changes in the structure of MPs. It was confirmed that the deterioration of the quality of frozen scallops was mainly caused by the structural degradation of MPs.

Furthermore, observations in different quadrants were related to the position of variables in the quadrant. For example, the best quality control sample had the highest values of Δ*H*_1_, α-helix, *T*_max1_, hardness, chewiness, springiness, Δ*H*_2_, *T*_max2_, and β-sheet as well as the low value of thawing loss, TVB-N, random coil, TBARS, MHC, actin, and β-turn. The comprehensive score (*P*) in different groups could comprehensively assess the degree of quality deterioration. *p* was calculated using the following formula: *P =* 0.595 × PC1 − 0.288 × PC2. The order of *P*-scores was control (1.46) < UIF-150 (0.37) < UIF-125 (0.06) < IF (–0.19) < UIF-175 (–0.24) < UIF-200 (–0.44) ≤ UIF-100 (–0.45) < AF (–0.55), indicating that the degree of quality deterioration of frozen stored AMS was UIF-150 < UIF-125 < IF < UIF-175 < UIF-200 ≤ UIF-100 < AF. UIF at 150 W was the best treatment condition to delay the quality deterioration of frozen scallops. The finding was consistent with that of the Taylor diagram analysis. Currently, studies on the level of UIF industrialization are very limited [23]. Scaling up laboratory-scale facilities to large-scale facilities suitable for industrial applications is always challenging [38]. This study provided a theoretical basis for the industrial application of ultrasonic-assisted freezing technology in the freezing and long-term storage of scallops.

## 4. Conclusions

AMS showed significant quality deterioration after 90 days of frozen storage. The structure degradation of MPs, fat oxidation, microbial growth, and texture deterioration during frozen storage contributed to the quality deterioration of AMS. Among them, the structural degradation of MPs was the most critical reason. The Taylor diagram and PCA results showed that UIF at 150 W was the best treatment condition to delay the quality deterioration of frozen scallops. The main reasons were summarized as follows: (1) The UIF-150 treatment effectively minimized the changes in the primary, secondary, and tertiary structures of MPs and protected the protein thermal stability of AMS by producing small and regular ice crystals in the AMS tissue during the freezing process. (2) The UIF-150 treatment retarded the fat oxidation and spoilage of frozen AMS by inactivating the corresponding enzymes and inhibiting the microbiological activities. (3) The aforementioned effects ensured that the AMS of the UIF-150 group had better microstructure, texture and quality after 90-day frozen storage. This study provided a theoretical basis for the industrial application of ultrasonic-assisted freezing technology in the freezing and long-term storage of scallops.

## Figures and Tables

**Figure 1 foods-11-03247-f001:**
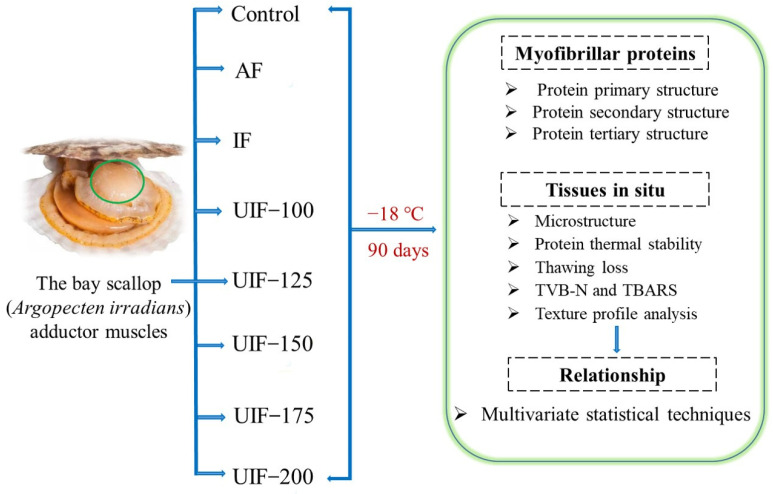
Procedure flow illustration for experiment design. Abbreviations are IF, immersion freezing; AF, air freezing; UIF-100, UIF-125, UIF-150, UIF-175, UIF-200, ultrasound-assisted immersion freezing at 100, 125, 150, 175, and 200 W, respectively; TVB-N, total volatile base nitrogen; and TBARS, thiobarbituric acid reactive substance.

**Figure 2 foods-11-03247-f002:**
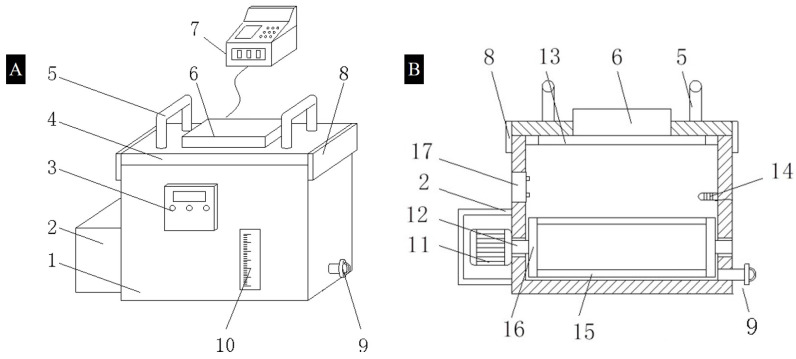
Schematic diagram (**A**) and box section view (**B**) of the ultrasound-assisted immersion freezing system (1. cabinet, 2. protective shell, 3. temperature control panel, 4. cover plate, 5. grip, 6. ultrasonic probe, 7. ultrasonic control panel, 8. limit magnetic plate, 9. drainpipe, 10. viewing frame, 11. motor, 12. connecting rod, 13. sealing ring, 14. temperature sensor, 15. lever, 16. rotating plate, 17. refrigeration unit).

**Figure 3 foods-11-03247-f003:**
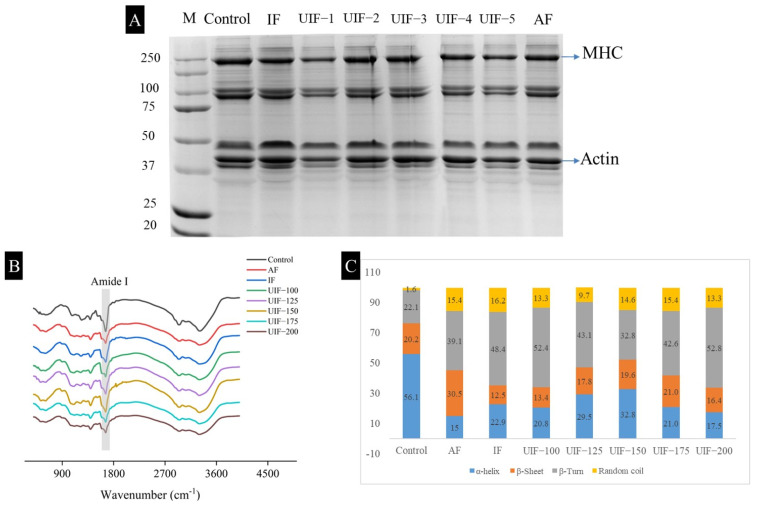
(**A**) SDS-PAGE pattern, (**B**) Fourier transform infrared spectra and (**C**) Secondary structure contents of myofibrillar proteins of control AMS and 90-day frozen stored AMS with different treatments. Abbreviations are M, protein molecular weight marker; UIF-1, UIF-2, UIF-3, UIF-4, UIF-5, ultrasound-assisted immersion freezing at 100, 125, 150, 175, and 200 W, respectively; and MHC, myosin heavy chain.

**Figure 4 foods-11-03247-f004:**
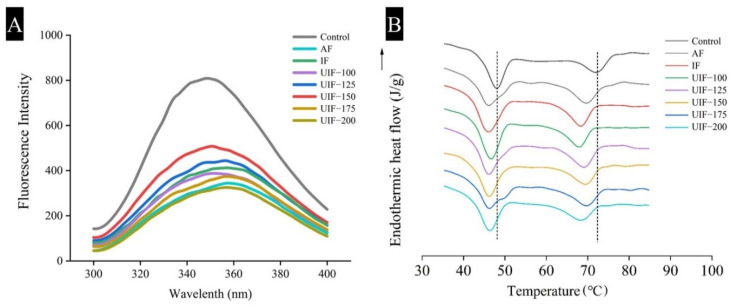
(**A**) Fluorescence images of myofibrillar proteins and (**B**) Thermal transition of control AMS and 90-day frozen stored AMS with different treatments. AF, air freezing; IF, immersion freezing; UIF, ultrasound-assisted immersion freezing at different ultrasound powers (100, 125, 150, 175, and 200 W).

**Figure 5 foods-11-03247-f005:**
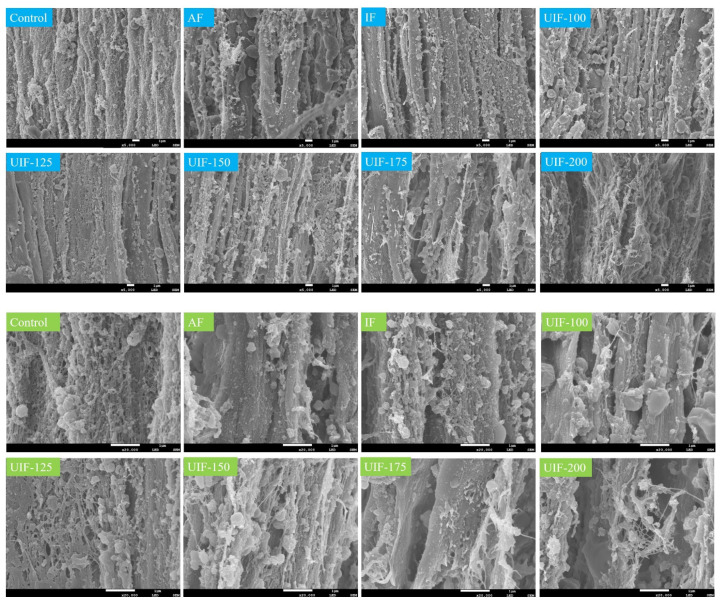
Scanning electron micrographs of the control AMS and 90-day frozen stored AMS with different treatments. AF, air freezing; IF, immersion freezing; UIF, ultrasound-assisted immersion freezing at different ultrasound powers (100, 125, 150, 175, and 200 W). The color of title is blue for magnification at 5000× and green for magnification at 20,000×.

**Figure 6 foods-11-03247-f006:**
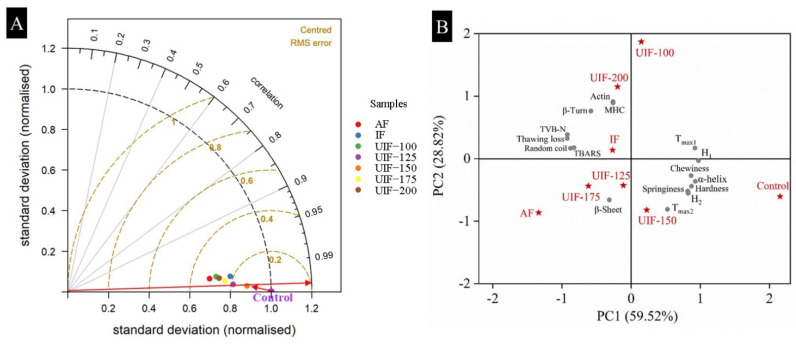
(**A**) Taylor diagram and (**B**) the principal component analysis (PCA) bi-plot of control AMS and 90-day frozen stored AMS with different treatments. AF, air freezing; IF, immersion freezing; UIF, ultrasound-assisted immersion freezing at different ultrasound powers (100, 125, 150, 175, and 200 W).

**Table 1 foods-11-03247-t001:** Changes in thiobarbituric acid reactive substance (TBARS), total volatile base nitrogen (TVB-N), thawing loss and instrumental texture of the control adductor muscle of scallop (AMS) and 90-day frozen stored AMS with different treatments.

	TBARS	TVB-N	Thawing Loss	Instrumental Texture Analysis		
				Hardness (g)	Springiness	Chewiness
Control	5.60 ± 0.08 ^e^	5.1 ± 0.3 ^e^	8.5 ± 0.11 ^h^	1449 ± 35 ^a^	0.66 ± 0.02 ^a^	540 ± 28 ^a^
AF	17.96 ± 0.66 ^a^	20.7 ± 0.5 ^a^	34.3 ± 0.13 ^a^	1034 ± 23 ^e^	0.52 ± 0.02 ^c^	343 ± 31 ^d^
IF	10.30 ± 0.31 ^cd^	17.9 ± 0.7 ^b^	32.2 ± 0.07 ^e^	1192 ± 24 ^c^	0.56 ± 0.02 ^bc^	354 ± 32 ^cd^
UIF-100	12.81 ± 0.86 ^b^	19.1 ± 0.5 ^ab^	33.2 ± 0.12 ^c^	1068 ± 23 ^e^	0.50 ± 0.03 ^c^	400 ± 22 ^cd^
UIF-125	9.95 ± 0.81 ^cd^	16.3 ± 0.7 ^c^	31.5 ± 0.08 ^f^	1190 ± 62 ^c^	0.54 ± 0.01 ^bc^	434 ± 75 ^bc^
UIF-150	8.35 ± 0.28 ^d^	14.3 ± 0.3 ^d^	26.1 ± 0.09 ^g^	1278 ± 31 ^b^	0.59 ± 0.06 ^b^	468 ± 61 ^ab^
UIF-175	10.39 ± 0.23 ^cd^	18.0 ± 0.4 ^b^	33.5 ± 0.14 ^b^	1139 ± 52 ^cd^	0.53 ± 0.03 ^bc^	389 ± 54 ^cd^
UIF-200	11.84 ± 0.40 ^bc^	19.9 ± 0.3 ^a^	32.6 ± 0.06 ^d^	1091 ± 43 ^de^	0.52 ± 0.02 ^c^	402 ± 31 ^cd^

Data are expressed as mean ± standard deviation, mean values in a column with different letters (a–h) are significantly different.

## Data Availability

The data presented in this study is contained within the article.

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
