# Peer review of "Effects of Ultrasound-Assisted Immersion Freezing on the Protein Structure, Physicochemical Properties and Muscle Quality of the Bay Scallop (Argopecten irradians) during Frozen Storage"

_foods, 2022, doi:10.3390/foods11203247_

Round 1
Reviewer 1 Report
Dear authors,
I list a few comments:
Linked by ultrasonic treatment, the ultrasonic interval of treatment is in a very close domain. Can you explain the reasons for choosing a set of cycle at 0,3? What is the advantage of that kind of treatment in your case? Please explain. The maximum intensities determine the diameter of the applied probe during UIF, please complete…
I am aware of the productivity of authors, but I suggest modifying and expanding the methods. For example, in 2.10. give us basic information, on what kind of probes are used in texture analysis, test speed...etc.
Figure 5. Please specify the meaning of the colors of applied process parameters, magnification at 5000x is blue, and 20000x is green.
Reviewer 2 Report
Congratulations on your work. A few parts of the manuscript needed to be improved.
1. The goals and principles of the techniques used in the work should be more clearly stated.
2. Figure 3B should be formatted as a table.
3. The data in Table 1 should be properly organized.
Reviewer 3 Report
Line 34: what’s referred as desired nutrition, might have to include the related nutrition details Line 40: what causes AMS to have poor taste/texture during freezing and frozen storage
Line 51: Needs further clarification on what exactly causes these changes
Line 54-59: Very vague
Line 65-68: Can be rewritten to effectively give more context, include all latest publications
Line 68: Please clearly present the hypothesis and contribution of your study in the Introduction section.
Line 70-80: The authors discuss well in this section, but there is no proper coherence, consequently, it should be rewritten. I ask authors to organize information. This version may make the reader confused. The final target should be clearly explained in the introduction.
Line 84: what was the temperature and how was it maintained
Line 85: needs further explanation on steps taken after receiving AMS
Line 89: experimental design is not clear, was it a randomized approach include all necessary experimental approach in text.
Line 105: how was this frequency and power level selected
Line 180: why did UIF150 behave differently, more explanation anticipated
Line 215: Further discussion is required
Line 221: interesting, please provide further discussion
Line 250: what does that indicate
Line 272: the scales and magnification were not visible; can you provide a note on this or provide it in fig caption
Line 281: Why was the UIF-150 treatment considered better
Line 319: the UIF-150 treatments retarded the texture deterioration, it was unclear what caused the lower thawing loss in the UIF-150 treatments
Line 335: UIF-150 group was deeded closest to the control, might have to support this claim better
Line 337: Being very wishful, please provide further clarifications from an industry standpoint
Line 420: Include a comment indicating the potential use of the proposed method at industrial level.
Round 2
Reviewer 3 Report
Thank you!